# Development of F-N-C-O Taguchi Method for Robust Measurement System Using a Case Study of T-Peel Test on Adhesion Strength

**Rozzeta Dolah** [1,*], **Mohamad Zaki Hassan** [1], **Santhana Krishnan** [2], **Faizir Ramlie** [1], **Mohd Fadhil Md Din** [2] **and Khairur Rijal Jamaludin** [1]

1 Department of Engineering, Razak Faculty of Technology and Informatics, Universiti Teknologi Malaysia, UTM KL, Jalan Sultan Yahya Petra, Kuala Lumpur 54100, Malaysia; mzaki.kl@utm.my (M.Z.H.); faizir.kl@utm.my (F.R.); khairur.kl@utm.my (K.R.J.)

2 Center of Environmental Sustainability and Water Security (IPASA), Research Institute of Sustainable Environment (RISE), Faculty of Engineering, Universiti Teknologi Malaysia, Johor Bahru 81310, Malaysia; kcsanthana@utm.my (S.K.); mfadhil@utm.my (M.F.M.D.)

* Correspondence: rozzeta.kl@utm.my

**Abstract:** A robust measurement system in the Taguchi Method as a testing method should be explained from the beginning of an experimental design until the application of the optimum condition. Measurement has always been described either by discussing the measurement concepts theoretically or demonstrating a case study on how the data measurement is being done practically. The distance between theory and practical case study that connects the test method used for measurement is always missing. In this paper, a case of T-peel test on strength measurement is used to reflect the robust measurement system, which includes the theory of experimental design together with methods to achieve the optimum condition. Seven control factors, two noise factors with one signal factor are used with orthogonal array L18. Not only the experiment results, but methodology on choosing the control, noise, and signal factors are described intensively. Therefore, the aims are to provide the procedure on evaluating optimum conditions, to analyze variability and optimization of T-peel test when measuring the strength, and to establish a mainstream flow to achieve high-quality experimental design for a robust measurement system. As a result, a robust measurement system that includes variation elimination is developed, which consists of four elements—F (function), N (noise), C (Control), and O (Optimization). The elements of the F-N-C-O system are connected to one another by the Plan-Do-Study-Act; P-D-S-A cycle. The results affect the existing measurement system by enlightening the black box of parameter design behind optimization results in Taguchi Method. Thus, the measurement is more convergent and obtain higher degree of confidence in parameter design.

**Keywords:** robust method; Taguchi parameter design; measurement system; signal-to-noise ratio; optimization

## 1. Introduction

The aim of a robust measurement system is to design a measurement system to get an optimized output by taking variation into consideration. Variation is the main keyword for robust engineering [1]. Robust engineering is described using parameter design in which a system is insensitive to variation. Thus, the development of a robust measurement system is important to obtain this objective. A robust measurement system reduces the variation that is contributed by the noise factor [1]. The noise factor cannot be eliminated, but the effect can be reduced by choosing the proper level for control factors.

This is done in parameter design that is used to improve the quality without removing the cause of variation and to make the product robust against the noise factors [1].

Measurement and instrumentation are the keys to enable the technology of science and other practical activities [2]. They reflect a very wide variety of equipment and techniques for a diversity of applications. Finkelstein [3,4] emphasized that measurement science should address the whole range of applications of measurement and provide a universal framework of concepts and principles for the applications of measurement. Dasgupta et al. [5] presents different variation countermeasure compared to Yano's [6] approach in metrology. Miller and Wu [7] also presented on different look at dynamic parameter design and robust design measurement systems. A performance measure on signal-response relationship is one of the contradictions.

Taguchi's strongest contribution to the design of experiments is to focus not only on minimizing the variability of the response, but also on optimizing the response. Thus, control factor and noise factor are separated in inner array and outer array, respectively. Most statistical methods concentrate on modeling, predicting, and controlling the average response, and this type of experimental design and analysis is crucial. Control factor is placed in inner array of orthogonal array [1]. Control factors affect process variability as measured by the signal-to-noise ratio (SNR). Control factor has three categories. The first category is the factor in which different settings give different average responses. These factors are said to be active but have no interaction with noise factors. The second category is a group of control factors that are active by virtue of having an interaction with noise factors [8]. They have the dispersion effect. Finally, the third category is the factors that have no effect on the response, called non-active. This kind of factor can be set at their cheapest or most convenient levels.

It is very important to have a robust measurement system explaining from the method of data measurement up to its optimization with minimum variation in the system. Most published papers discussed the test method used in some case studies [9–21] while others discussed the measurement concepts and theories [8,10,22–25]. However, less research has been done to connect both aspects in a single study [9]. Thus, the main study in this paper connects the test method used for measurement and the theories behind the setting up of its measurement which are described.

A case of T-peel strength measurement is used to depict the robust measurement system for parameter design of Taguchi Method. This paper aims to provide the procedure on determining optimum conditions to analyze variability and optimization when designing a measurement system, and to establish a process flow to achieve high-quality experimental design for a robust measurement system [26]. Therefore, a systematic measurement system is developed that results in a robust product and process by using parameter design. The finding of this paper reveals the quality result or optimum condition in a measurement system which consists of the function, noise, control parameter, and optimization elements stated as F-N-C-O.

## 2. State-of-the-Art Methodology

### 2.1. The Need of Revised Standard

A practical experiment on T-peel test measurement is done on the flexible packaging film as a method for measuring the peel strength of an adhesive. Peel tests are most commonly used to evaluate laminated film or bonded adhesives. Thus, a peel test is preferred when working with multiple film packaging in this study, namely polyethylene (PET), polyamide, aluminum (Al), cast polypropylene (CPP), and bonded with adhesives. There are four main types of peel tests: 90° peel, 180° peel, T-peel, and climbing drum peel. The 90° peel test is suitable for a flexible adhesive material that is adhered to a more rigid substrate. The 180° peel test is best used when the flexible substrate can be bent back by 180°. The T-peel test is best used when both adhesive and adhered are similar or flexible. Therefore, the T-peel test is the most suitable peel test to measure the peel strength of this material. The T-peel test apparatus as described in Figure 1 is a newly revised apparatus derived from current standard JISK

6854-3 (2008) [27] and American Society of Testing and Materials; ASTM 1876-08 [28] to fulfill industrial requirement. The apparatus in Figure 1 is used to develop the optimum condition for flexible film.

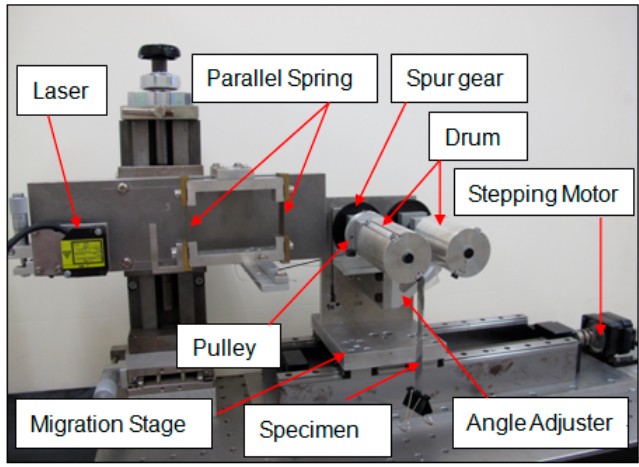

**Figure 1.** T-peel test apparatus set-up (1 cm photo:5 cm actual).

As shown in Figure 2b, in previous standardized method JISK 6854-3 (2008) [27] and American Society of Testing and Materials; ASTM 1876-08 [28], the peel angle is not maintained at 90° for flexible film, thus it leads to a big variation in peel strength during peeling. Therefore, the new testing apparatus is developed shown in Figure 2c to encounter the variation problem in flexible film due to peel angle instability.

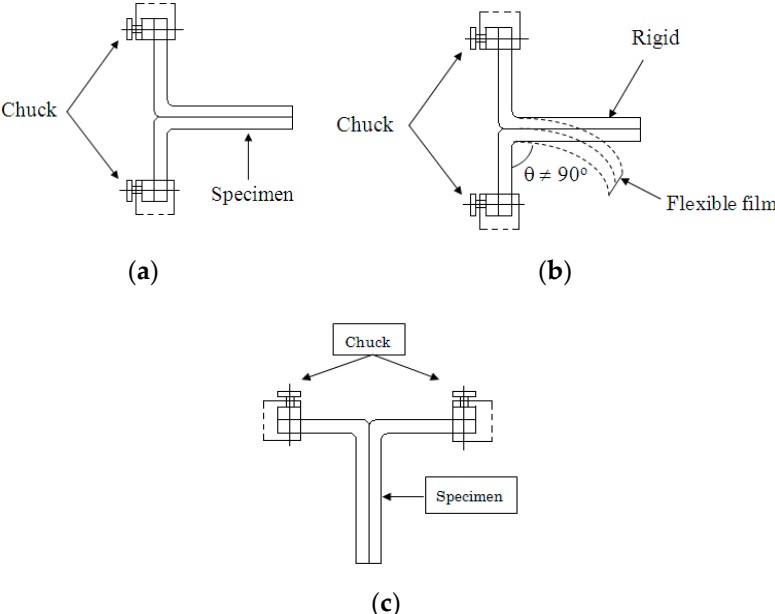

**Figure 2.** (**a**) Standardized T-peel test method Japanese Industrial Standard (JIS) [27] and American Society of Testing and Materials (ASTM) [28] and (**b**) Failure of flexible film specimen to stabilize the peel angle, (**c**) the new testing apparatus for T-peel test as shown in Figure 1.

The drum rotates by the spur gear according to the peel speed as a string is attached at a fixed point and tied on the drum's pulley as shown in Figure 3 schematic diagram. As the drum rotates according to the peel speed, the migration stage is being moved back and forth to the initial state. Angle adjuster is used to change the peel angle according to several setting from 0° to 180°. The flexible film adhesive is attached to the drum. A weight (paper clip) was fixed on the free end of the film to

keep the specimen in T-shape. The drum rotates according to peel speed as a string is attached at a fixed point and tied on the drum's pulley. Peel speed and peeling distance can be changed according to apparatus specifications. A parallel spring is pulled by pulley wire along the peeling process. Three spring thicknesses were used for this study: 0.3 mm, 0.4 mm, and 0.5 mm. During the peeling process, displacement is triggered by a parallel spring caused by peel strength and detected by a laser sensor. This apparatus can obtain a wide range of peel strength measurements by changing the spring thickness.

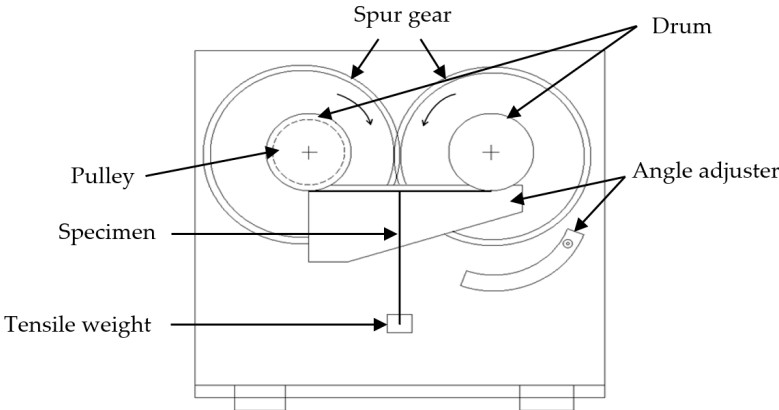

**Figure 3.** T-peel test apparatus schematic diagram.

Peel strength increased proportionally to specimen width as shown in Figure 4. Higher strength is needed to peel away the adherend from the adhesive as the specimen width increases.

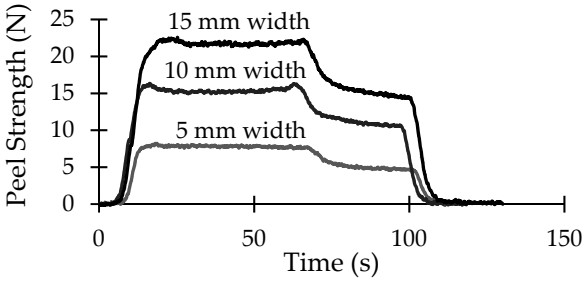

**Figure 4.** Peel strength increased proportionally to specimen width.

### 2.2. P-Diagram of T-Peel Test

Parameter diagram or P-diagram is constructed to understand all parameters involved in a system as shown in Figure 5. The function of Al-CPP T-peel test is to measure adhesion peel strength in flexible film. Thus, the response or output of T-peel test is peel strength, which is measured in Newton (N). The input of T-peel test is known as signal factor. In the ideal function, the energy transformation occurs for three different specimen widths that are 5 mm, 10 mm, and 15 mm. Signal factor, in this study, is specimen width that is a controllable variable to actualize the intention (variation in peel strength) to achieve robust condition regardless of various width condition [26].

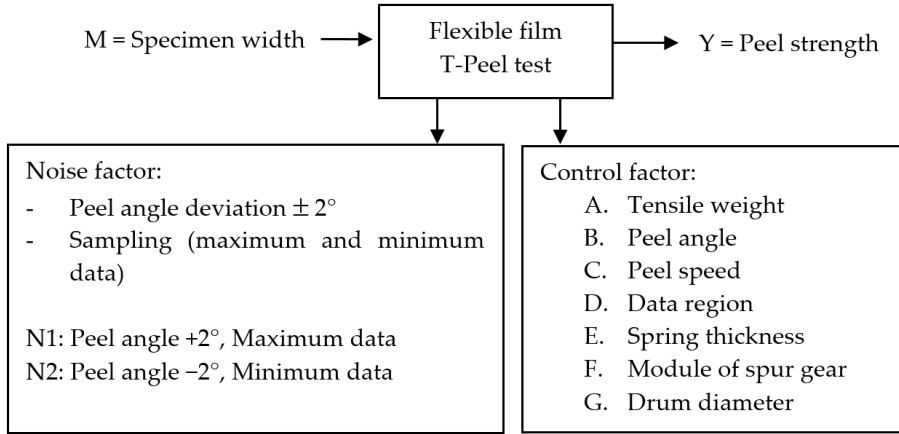

**Figure 5.** P-diagram for flexible film.

In P-diagram, robustness is optimized by evaluating the control factors and their levels. The noise factor condition is varied accordingly to minimize variation that influences the response. Signal-to-noise ratio (SNR) with dynamic response (Equation (1) is used in this study due to the signal factor existence. A dynamic signal-to-noise ratio (SNR) has been used in this study where the specimen widths of 5 mm, 10 mm, and 15 mm as the signal factor levels are used to measure the peel strength linearity.

*S/N* ratio,

$$\eta = 10 \log (1/r) [(S_\beta - V_e)/V_N] \tag{1}$$

where $S_\beta$ = variation caused by the linear effect, $V_e$ and $V_N$ = error variance (error variance/DOF), *r* = total number of measurements under signal *(r* is also the effective divisor due to level changes of signal factor).

Noise factor is an uncontrolled factor during normal production or use but is controlled during the experiment. There are three types of noise: outer noise, which is caused by environmental conditions; inner noise, which is caused by the deterioration of elements or materials in the product; and between-product noise, which is caused by piece-to-piece variations between products. Noise factors are likely to produce variability in the response. For noise factor, historical data have proven that the peel angle would vary during exchanging the peel angle setting and during peeling process [2,28,29]. Peel angle deviation will affect the peel strength; thus, peel angle is considered as a source of variability. Two noise factors are considered in this study: peel angle deviation ±2° and tensile weight *w*. Peel angle is adjusted to three levels: 60°, 90°, and 120°. Peel angle ±2° is a noise factor because it is possible to have an inaccurate reading if the peel angle is changed by the angle adjuster by two degrees due to natural movement, as shown in Figure 6. The noise in peel angle is defined as deviation ± 2° for each level. Noise 1 is the higher level (N1 = +2° and maximum peel strength value) and Noise 2 is the lower level (N2 = −2° and minimum peel strength value). The plot for noise factor is shown in Figure 7. After completing the noise strategy, the selection of a control factor is done. The objective of this T-peel test is to satisfy the industry requirement of getting the minimum variation for flexible film.

Thus, control factors are selected that may affect variability in the response, and possibly the mean of the response. The controllable factors or inner array are chosen based on testing and design conditions that can affect the variance. The controllable factor selection is based on the historical data of experiment results, preliminary tests, theories and available knowledge, and expert's opinion.

The controllable factors are considered from testing condition and design of apparatus condition.

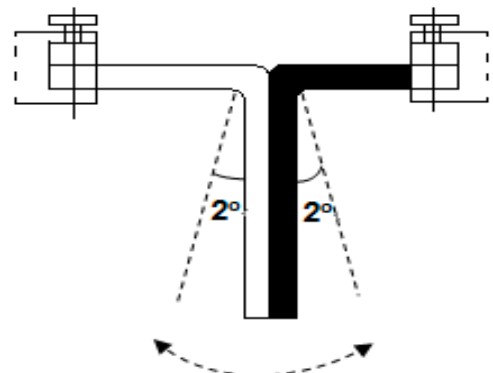

**Figure 6.** Deviation in peel angle during T-peel.

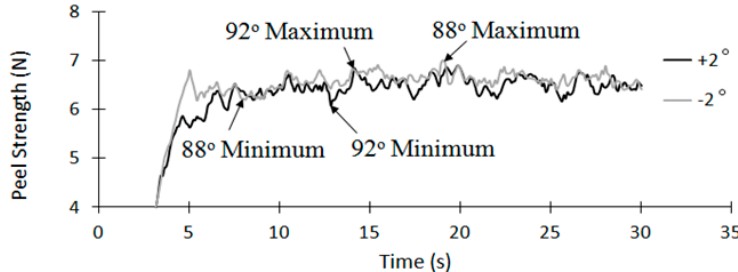

**Figure 7.** N1 and N2 plot for noise factor.

The control factor from testing condition includes tensile weight that is used to keep the specimen in T-shape, peel angle, peel speed, and peeling curve region. From the design of apparatus condition, the control factors are parallel spring thickness, module of spur gears, and drum diameter. The factor's level is decided based on the experiments' objective. The levels must not be so close to each other that the effect on the response is not observable or undetected. Levels must also not be very far apart that there is a region of unknown process behavior. Previous process knowledge is useful to determine the level. For example, three levels are chosen to observe the curvature effect on the response. Based on Figure 3, control factors B, C, D, E, F, and G are analyzed with three levels. Two levels are chosen to determine whether the factor has an effect on the response. Control factor A tensile weight is analyzed with two levels 4 g and 8 g. More than three levels are suitable to observe significant trend or behavior, such as sudden rise or drop at certain levels. The experimental design space is large, and it needs a strategy to explore. After determining the control factors and factor's level, they are assigned into an orthogonal array. An orthogonal array is used for optimization to maximize the signal-to-noise ratio [1,8,9]. Balance set of experimentation runs is provided by orthogonal array. Design of experiments using orthogonal array $L_{18}$ is utilized with one two-level factor (tensile weight) and six three-level factors (peel angle, peel speed, data region, spring thickness, module of spur gear, and drum diameter) as shown in Table 1.

The optimization approach starts with the research motivation to establish a procedure on T-peel test optimization followed by a determination of optimum conditions. An experimental confirmation test was performed to validate the estimated condition. The experience from $L_{18}$ in selecting control and noise level is presented and needs to be carefully done. Ideally, a system with zero or minimum noise is desired. After optimization, the gap between noise level N1 and N2 must be as small as possible to produce an ideal function, shown in Figure 8.

**Table 1.** Experimental setup.

| Control Factor | Unit | Level 1 | Level 2 | Level 3 |
|---|---|---|---|---|
| A: Tensile weight | g | 4 | 8 | |
| B: Peel angle | ° | 60 | 90 | 120 |
| C: Peel speed | mm/s | 6 | 9 | 12 |
| D: Data region | % | 30 | 50 | 70 |
| E: Spring thickness | mm | 0.3 | 0.4 | 0.5 |
| F: Module of spur gear | d/z | 0.5 | 1 | 2 |
| G: Drum diameter | mm | 20 | 30 | 40 |
| *Signal Factor* | | *Levels* | | |
| M: Specimen width | mm | 5 | 10 | 15 |
| *Noise Factor* | | *Level N1* | *Level N2* | |
| Peel angle | θ | 2 | −2 | |
| Peel strength sampling | N | Maximum | Minimum | |

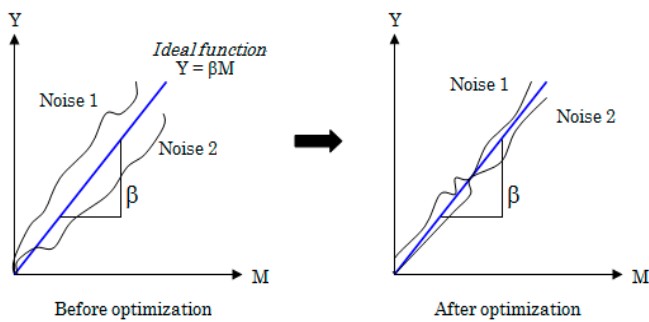

**Figure 8.** Variability improvement after optimization.

The robust design engineering methodology is developed and briefly described as follows:

*Step 1:* Enable functionality of the system. Carefully analyze the ideal function that transforms the energy into quality characteristic. Construct P-diagram to get a whole picture of the system.

*Step 2:* Identify the problem by selecting the response based on experiment's objective. The response may be maximized, minimized, or taken to a target value. The mean and variance of a response can be studied simultaneously. Construct an ideal function and P-diagram. Determine the input (signal factor) and output (response) of the experiment.

*Step 3:* Select noise factor and level for outer array. Relate with response objective, for example, if the objective is to minimize variation of peel strength, make sure the noise factor can produce the variation in peel strength and the design space is covered as best as it can.

*Step 4:* Select control factor and level for inner array. Consideration of factor level must in line with objective or intended effect on the response such as curvature, effect presence, and other behavior or trend.

*Step 5:* Construct an orthogonal array based on number of factors and levels. Implement the experiment. SNR and sensitivity response plot are analyzed.

*Step 6:* Check on reproducibility. Estimation and confirmation dB gain is compared. Rule of thumb: less than 3 dB gain difference is good.

*Step 7:* If the intention is to move the mean to target, perform adjustment step. If there is no intention to move the mean to certain target, steps 1 to 6 are sufficient enough.

Identifying the experiment's objective is crucial because it can affect the selection of noise and control factors. General guidelines are described step-by-step from selecting the response up to decision making on the optimum dB gain.

The steps explained from Step 1 to Step 7 can be summarized in a closed-loop measurement system mainly in four sub-systems; F for Function system, N for Noise system, C for Control system,

and O for Optimization system. The application of the optimum condition called robust condition is the output of the F-N-C-O ladder. The details of each steps are described in Section 4 (Sections 4.1–4.4). Discussion section explaining the results gained from Section 3. Experimental Results.

## 3. Experimental Results

The peel strength result is measured to calculate the robustness metric, signal-to-noise ratio SNR, $\eta$, and sensitivity, $\beta$. In a measurement, $V_e$ represents the correction of error variance that means the variation of the data measured in a sample. $V_e$ reflects the variation that affects the accuracy and precision of a measurement in a controlled condition. In a robust measurement system, $V_e$ is calculated with consideration of $V_N$, the variation of compounded noise factor. This represents the metric for robustness called signal-to-noise ratio, $\eta$ in decibel unit, dB.

SNR example of calculation is shown below by taking the result in Table 2 for run 1:

$$\text{SNR, } \eta = 10 \log ((1/r \cdot r_o) (S_\beta - V_e)/V_N) \tag{2}$$

$$S_\beta = 542.82 = ((4.45 + 4.53)5 + (8.51 + 8.97)10 + (13.51 + 12.94)5)^2/2(5^2 + 10^2 + 15^2)$$

$$V_e = S_e/f_e = (S_T - S_\beta - S_{Nx\beta})/4 = 0.0689, f_e = 4 \tag{3}$$

$$S_T = 4.45^2 + 4.53^2 + 8.51^2 + 8.97^2 + 13.51^2 + 12.94^2 = 543.12$$

$$S_{Nx\beta} = 0.0180 = ((4.45)5 + (8.51)10 + (13.51)15)^2 + ((4.53)5 + (8.97)10 + (12.94)15)^2)/(5^2 + 10^2 + 15^2) - S_\beta$$

$$V_N = S_e'/f_e' = (S_T - S_\beta)/5 = 0.05879, f_e' = 5$$

Signal-to-noise ratio; SNR,

$$\eta = 10 \log_{10}(1/2(5^2 + 10^2 + 15^2))[(S_\beta - V_e)/V_N] = 11.20 \text{ dB}$$

Sensitivity,

$$\beta = 10 \log(1/r \cdot r_o) (S_\beta - V_e) = 10 \log_{10}(1/2(5^2 + 10^2 + 15^2))(S_\beta - V_e) = -1.10 \text{ dB} \tag{4}$$

**Table 2.** $L_{18}$ experimental result of signal-to-noise ratio (SNR) and sensitivity $\beta$.

| Run # | A | B | C | D | E | F | G | 5 mm N1 | 5 mm N2 | 10 mm N1 | 10 mm N2 | 15 mm N1 | 15 mm N2 | SNR | Sensitivity |
|---|---|---|---|---|---|---|---|---|---|---|---|---|---|---|---|
| 1 | 1 | 1 | 1 | 1 | 1 | 1 | 1 | 4.45 | 4.53 | 8.51 | 8.97 | 13.51 | 12.94 | 11.20 | −1.10 |
| 2 | 1 | 1 | 2 | 2 | 2 | 2 | 2 | 6.31 | 6.12 | 12.94 | 11.95 | 18.83 | 18.15 | 10.06 | 1.85 |
| 3 | 1 | 1 | 3 | 3 | 3 | 3 | 3 | 8.79 | 8.46 | 16.96 | 16.43 | 24.57 | 23.88 | 10.27 | 4.29 |
| 4 | 1 | 2 | 1 | 1 | 2 | 2 | 3 | 8.39 | 8.08 | 16.50 | 15.72 | 24.08 | 23.36 | 11.55 | 4.05 |
| 5 | 1 | 2 | 2 | 2 | 3 | 3 | 1 | 3.94 | 3.57 | 7.98 | 7.50 | 11.53 | 10.82 | 7.12 | −2.46 |
| 6 | 1 | 2 | 3 | 3 | 1 | 1 | 2 | 7.20 | 7.05 | 13.81 | 13.52 | 19.73 | 18.91 | 6.76 | 2.41 |
| 7 | 1 | 3 | 1 | 2 | 1 | 3 | 2 | 6.38 | 6.38 | 13.28 | 12.74 | 18.97 | 17.83 | 6.91 | 1.95 |
| 8 | 1 | 3 | 2 | 3 | 2 | 1 | 3 | 7.97 | 7.42 | 16.27 | 15.67 | 24.76 | 23.60 | 10.11 | 4.10 |
| 9 | 1 | 3 | 3 | 1 | 3 | 2 | 1 | 3.93 | 3.75 | 7.51 | 7.28 | 11.33 | 11.10 | 14.44 | −2.53 |
| 10 | 2 | 1 | 1 | 3 | 3 | 2 | 2 | 6.02 | 5.19 | 12.03 | 11.19 | 16.91 | 16.60 | 8.17 | 1.07 |
| 11 | 2 | 1 | 2 | 1 | 1 | 3 | 3 | 9.79 | 8.95 | 17.92 | 17.28 | 26.98 | 25.29 | 7.46 | 4.89 |
| 12 | 2 | 1 | 3 | 2 | 2 | 1 | 1 | 4.34 | 4.10 | 8.34 | 7.97 | 12.17 | 11.91 | 11.74 | −1.84 |
| 13 | 2 | 2 | 1 | 2 | 3 | 1 | 3 | 7.75 | 7.23 | 15.14 | 14.34 | 22.10 | 21.12 | 9.52 | 3.25 |
| 14 | 2 | 2 | 2 | 3 | 1 | 2 | 1 | 4.74 | 4.44 | 9.17 | 8.81 | 13.38 | 12.90 | 10.44 | −1.06 |
| 15 | 2 | 2 | 3 | 1 | 2 | 3 | 2 | 6.77 | 6.23 | 12.82 | 12.51 | 19.25 | 18.54 | 11.87 | 2.04 |
| 16 | 2 | 3 | 1 | 3 | 2 | 3 | 1 | 4.04 | 3.22 | 7.74 | 6.92 | 11.64 | 11.13 | 4.84 | −2.51 |
| 17 | 2 | 3 | 2 | 1 | 3 | 1 | 2 | 5.95 | 5.52 | 12.22 | 11.56 | 17.84 | 17.09 | 9.90 | 1.36 |
| 18 | 2 | 3 | 3 | 2 | 1 | 2 | 3 | 8.94 | 8.73 | 16.32 | 16.01 | 24.69 | 25.03 | 11.13 | 4.37 |

Table 2 shows the raw data of peel strength measurement with calculated SNR and sensitivity $\beta$ using dynamic Taguchi method. In $L_{18}$ as shown in Table 2 orthogonal array, there are 108 observations implied (18 runs × 3 signal level × 2 noise level): The raw data are positioned under outer array below the signal and noise factor column.

Optimum condition is determined from the highest peak of SNR and maximum SNR explained the power of signal is larger than power of noise, which means the variability is small. Thus, the optimum condition is influenced by noise factors. Figure 9 shows the L18 (a) SNR plot and (b) Sensitivity plot.

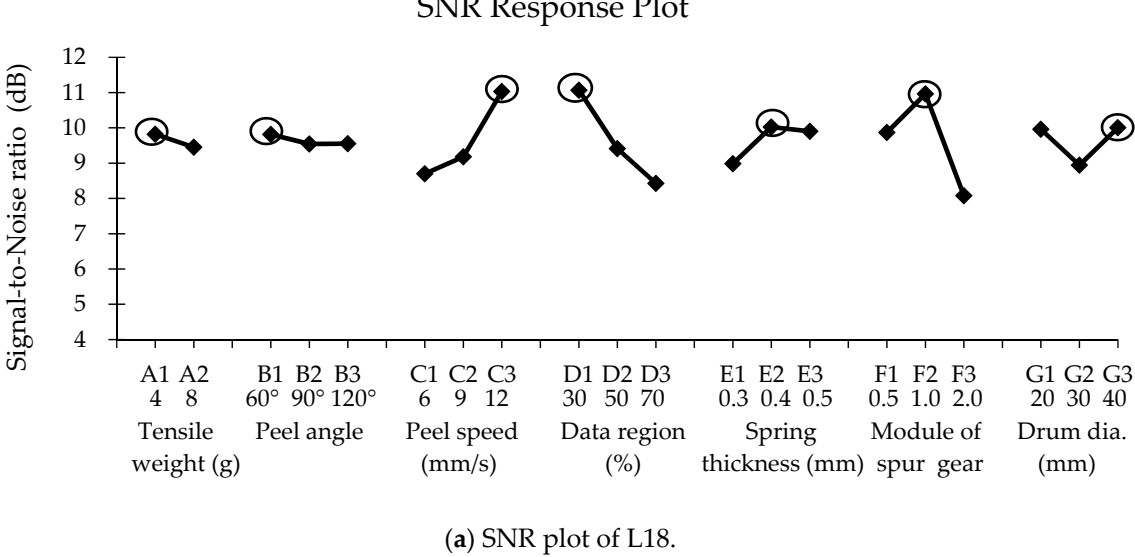

(**a**) SNR plot of L18.

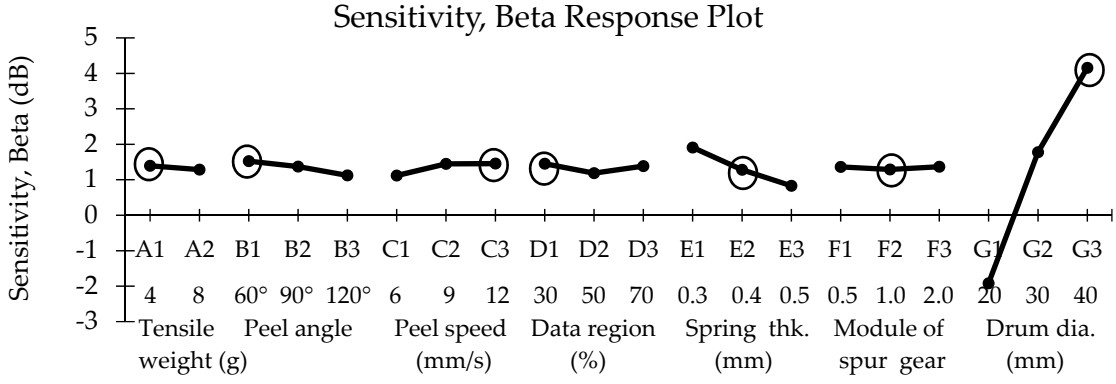

(**b**) Sensitivity, $\beta$ Plot of L18.

**Figure 9.** Result plot for SNR (**a**) and sensitivity, $\beta$ (**b**).

The final step is to predict and verify the improvement in peel strength variation using the optimum level in SNR response plot. Optimum condition is taken from the highest peak of SNR plot that is A1 B1 C3 D1 E2 F2 G3. The worst condition is taken from the lowest SNR peak from each factor that is A2 B2 C1 D3 E1 F3 G2. The effect of the optimum condition is shown by the dB gain size between optimum and worst SNR. A confirmation run is done to check the SNR reproducibility of the estimation and confirmation experiment. The optimum condition based on SNR plot is shown in Table 3.

Based on Table 3, the ideal function plot of optimization step shown in Figure 10 described the dB gained between optimum and worst conditions.

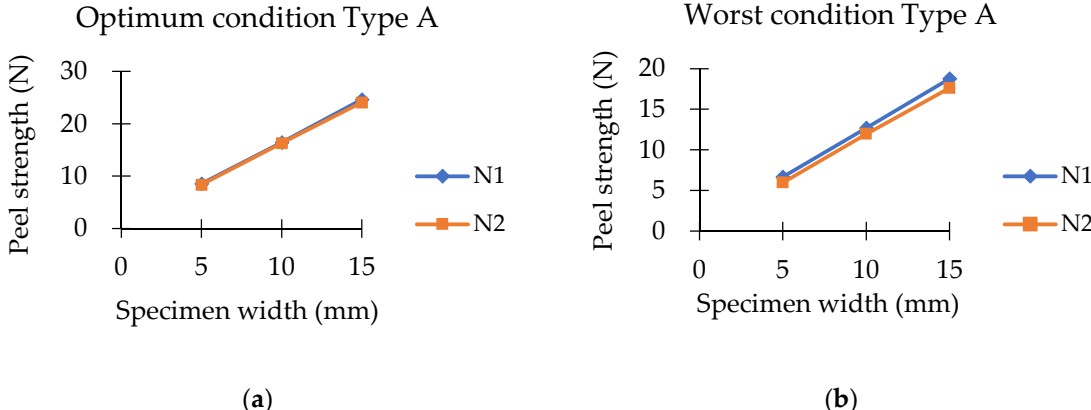

**Figure 10.** Ideal function plot for optimum (**a**) and worst condition (**b**).

**Table 3.** Optimum and worst condition with confirmation run.

| Type A | Condition | Estimated SNR (dB) | Confirmation SNR (dB) |
|---|---|---|---|
| Optimum | A1 B1 C3 D1 E2 F2 G3 | 14.91 | 14.82 |
| Worst | A2 B2 C1 D3 E1 F3 G2 | 4.3 | 7.07 |
| SNR dB Gain | | 10.61 | 7.75 |

## 4. Discussions

The findings can be depicted as one measurement system design for parameter design in the Taguchi method. The literature review explained that most Taguchi method applications only informed about application of the tool without discussing the concept of variation measurement in the early implementation [29–33]. The parameter design in a robust engineering tool is further analyzed to ensure the measurement covers total variability in data. There are four stages in measurement systems for this flexible film. The first measurement system is the function system measurement (F). Secondly, noise strategy measurement (N) is done followed by control factor selection (C) and optimization (O). Figure 11 showed the measurement system in parameter design.

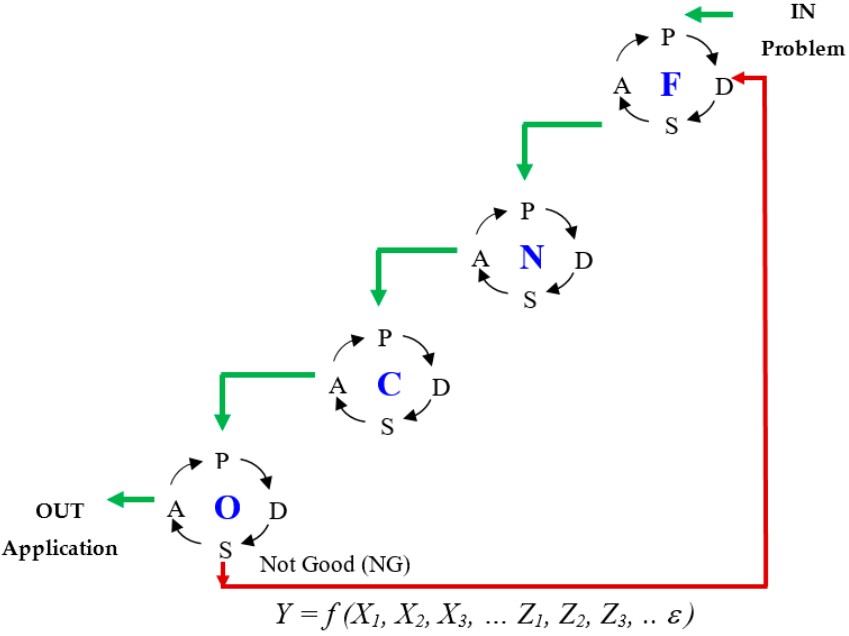

**Figure 11.** F-N-C-O ladder for measurement system in robust design engineering.

The system is a close-loop measurement system called an F-N-C-O ladder. The problem statement is the system ignition followed by the four sub-systems; F for Function system, N for Noise system, C for Control system, and O for Optimization system. The application of the optimum condition is the output of the F-N-C-O ladder. The elements F-N-C-O are connected to each other by the Plan-Do-Study-Act so-called P-D-S-A cycle. In each element, the P-D-S-A cycle is used for continuous improvement from one sub-system to another. Each sub-system starts with P: Plan stage and ends up with A: Act stage. The Act stage in Function system (F) moves the Plan stage in Noise system (N).

Act stage in N moves the Plan stage in Control system (C) and so on. In optimization system (O), S: Study stage analyzes the normality of the data spread. If the spread of data is not good (NG), the flow is back to Function system (F) under D: Do stage of selecting the quality characteristic. If the reproducibility is low detected in O system, the experiment is a failure. The failure is often related to basic function setting, quality characteristics, and other components of measurement. Thus, the flow is back to F. This P-D-S-A cycle ensures the mobility of the system and dependency to each sub-system. Details on each element are elaborated in further sections.

Bo Bergman and Bengt Klefjo [34] explained that a robust design is considered as improvement stage of the product development process. A simple mathematic description is illustrated with an output variable denoted as y with a target value $y_o$. Three design parameters $X_1$, $X_2$, $X_3$ and a noise factor $Z$ as an uncontrollable variation is shown in Equation (5):

$$Y = f(X_1, X_2, X_3, \dots Z_1, Z_2, Z_3, \dots \varepsilon) = b_o + b_1 X_1 + b_2 X_2 + b_3 X_{3+} b_Z Z_+ b_{2Z} X_2 Z_+ \varepsilon \tag{5}$$

$\varepsilon$ is an unknown, small residual term which is independent on the design parameter.

With $X_2 = -bZ/b_2Z$, the influence of noise factor $Z$ is completely disappearing. The equation is said to be robust because the influence of noise factor is minimized and disappeared. Therefore, the equation is placed at the bottom as the F-N-C-O foundation. The noise factors in an experiment can be varied, assumed that it is controllable during the experiment and uncontrollable in real life. There is some case whereby noise is not detected and its appearance is not even known. These variations are caused by extraneous factors. Extraneous factor could deviate the output result unintentionally.

### 4.1. Function System (F)

This stage consists of enabling the function, energy transformation, quality characteristic selection, and finally forming the ideal function. In order to enable the functionality of the system, a careful analysis of the ideal function that transforms the energy into quality characteristic is done. This is done in "Plan" stage to define the function of the system that is being investigated. The response, Y, or also called quality characteristic represents the energy transformation of the system, then the interactions are greatly reduced because energy is additive. In order to improve quality, it is important not to measure and analyze the response, but to measure the function, that is energy-related because energy has additivity.

In this paper, peel strength is the energy for flexible film testing. At "Do" stage, peel strength has been used as quality characteristic for the practical experiment using parameter design. After the quality characteristic is identified, the type of signal-to-noise ratio (SNR) is studied. In "Study" stage, analysis on which suitable quality characteristic is done to ensure the variation is totally captured. It is important to identify whether or not the system has the signal factor. There are static SNR without signal factor and dynamic SNR with signal factor. Static SNR consists of bigger-the-better (BTB), smaller-the-better (STB), nominal-the-best (NTB), and operating window (OW). In this research, dynamic SNR is chosen because of signal factor that is specimen width used.

Next, under "Act" stage, an ideal function and finally P-diagram is constructed to get a full view of the parameter design system. An ideal function shows a relationship between a signal and an output characteristic under certain conditions of use. In robust engineering, research and development are conducted by reducing the variability of a function under various conditions and bring the function as

close as possible to the ideal function under standard condition [5,6]. The detail of Function element is shown in Figure 12.

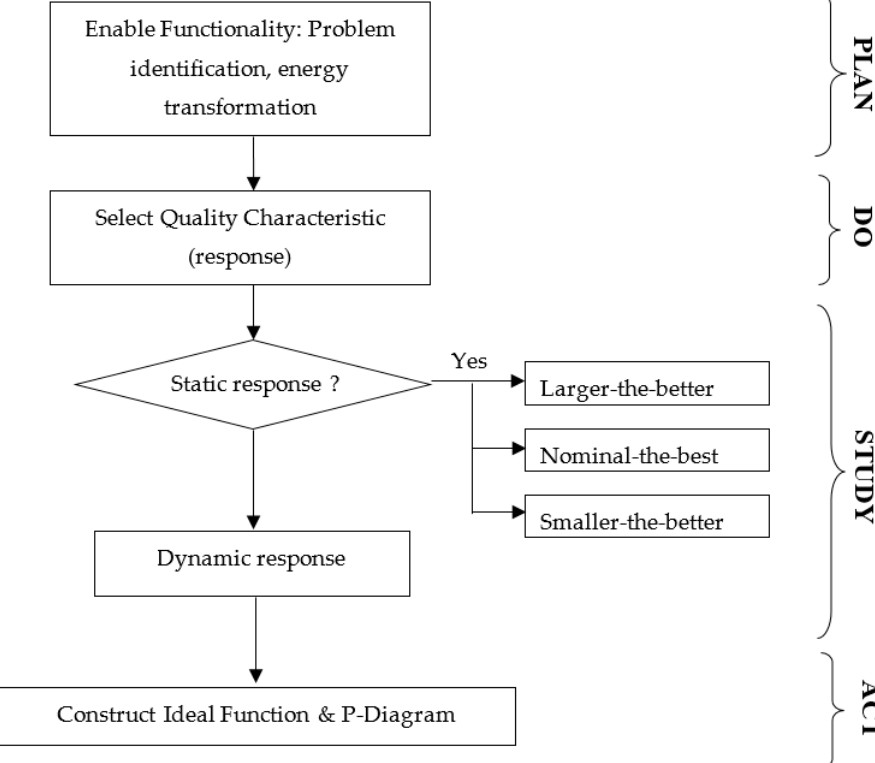

**Figure 12.** Function (F) sub-system of F-N-C-O.

*4.2. Noise System (N)*

Noise factors cannot be controlled during normal production or use. Thus, noise factor is likely to produce variability in response. In other words, noise is a variable that affects product functions. Three types of noise are outer noise caused by environmental conditions, inner noise caused by deterioration of elements or materials in the product, and between-product noise caused by piece-to-piece variation between products [5,6]. A robust design experiment searches for values of the control factors that can be controlled during production and it is important to make the product or process insensitive to changes in noise factors.For a robust design experiment, start by selecting the response, and then choose noise factors that are likely to produce variability in response, and finally select the control factors that are likely to affect the variability and the mean of the response. After identifying the experiment objectives, it is usually preferable to select the responses before selecting the noise factor. Therefore, the "Act" stage of constructed ideal function and P-diagram in F sub-system is connected to the "Plan" stage in N sub-system. After function measurement is determined for the quality characteristic, noise measurement is done. When planning an experimental design, selecting factors that affect the response and their levels of value or setting are very important. If incorrect factors and levels are chosen in the experiment, the results may be incomplete or misleading. A number of factors and levels are chosen based on the objectives of the experiment such as smaller-the-better, bigger-the-better, or nominal-the-best. Engineering knowledge of the process can be used to select noise factors and levels. Historical data, previous experimental results, theoretical knowledge, expert opinions, observational data, and other relevant data can be used in judging what the noise factor should be. The noise strategy is very important because it involves the selection of noise factor. Engineering knowledge is very useful in ensuring the stability of the optimum condition in long term. When the source of variation is identified clearly, this will minimize the risk of having an unstable

optimum condition. The robustness of the measurement result is heavily dependent on the noise factor. In the long term, reproducibility of measurement must be equal or almost equal to the measurand.

Figure 13 describes the important flow for Noise (N) system strategy. In selecting the noise level, the range of factor levels should be selected as the levels are not so close to each other because the effect on the response is not observable or important nearby effects will be undetected. The level also should not be so far apart that there is a region of unknown process behavior between the factor levels. The level also depends on the response being considered. Two-level is chosen when the factor has a linear effect on the response. Three-level of a factor is chosen to study curvature in the response. Normally in two-level factor, it is possible to assess whether there is curvature in one or more factors by adding center points at the center of its range. Four-level factor or more is to study further curvature to locate sudden rise or drop in the response. In other words, extra level is meant to understand some patterns or change behavior. Then, the noise validation study is made in "Do" stage to categorize the group of noise parameters. For example, set 1, or usually denoted as N1, gathers the low setting of noise parameters. Set 2, or N2, denotes the high setting of noise parameters. This is the special characterization of the Taguchi method that the noise parameter is separated in outer array. The effect of noise parameter is studied in the Taguchi method and placed in outer array. The Taguchi method focuses on achieving robustness in functional performance. However, in classical design of experiments (DOE), the objective is to minimize the effect of parameters using blocking or randomization strategies. Thus, in DOE all parameters are placed in one array and have no distinction of control and noise parameters.

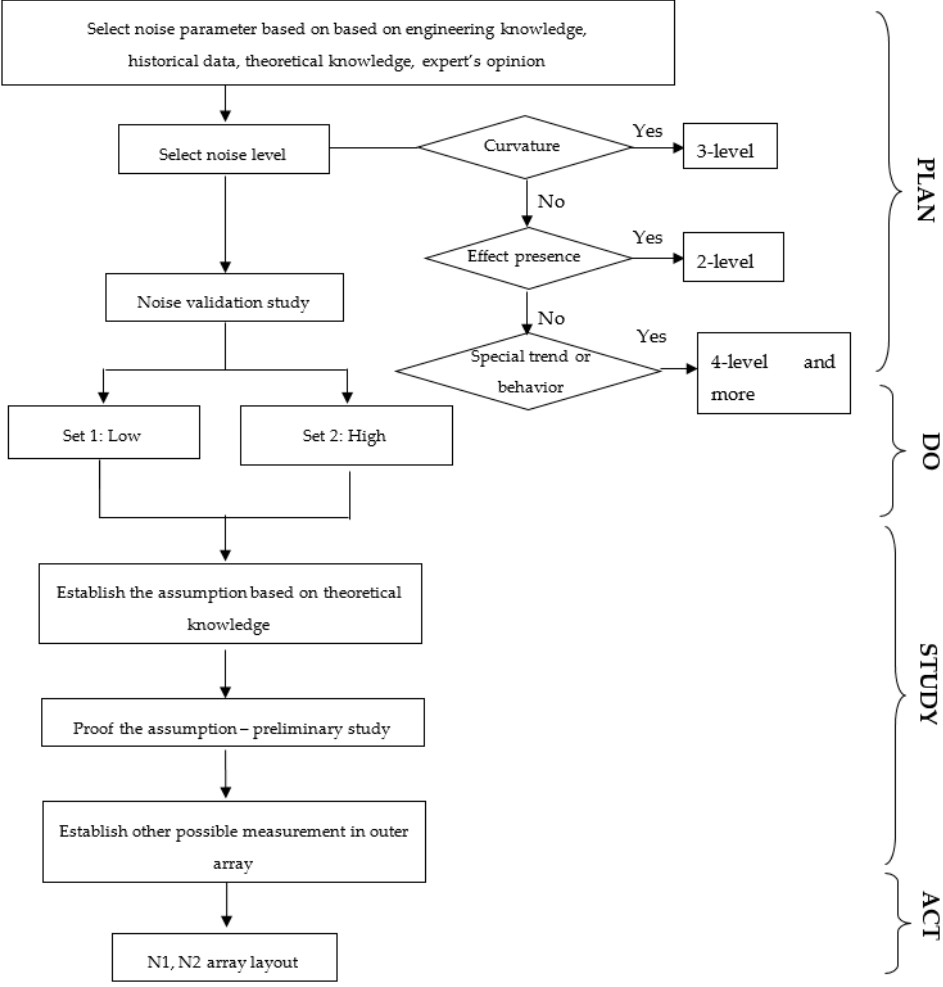

**Figure 13.** Noise (N) sub-system of F-N-C-O.

### 4.3. Control System (C)

Selecting the control parameters is started with response's objective, sometimes called its quality characteristic. Similarly, in selecting the noise parameter, engineering knowledge of the process can be used to select and judge the control parameters and their level apart from historical data, previous experimental results, theoretical knowledge, expert opinions, observational data, and other relevant data. After identifying the noise factor, control factor evaluation is done. Noise measurement system is done before the control factor system to ensure the effect of all the noise parameters can be minimized by the best control-factor-level combination. The final control factors for the T-peel test are tensile weight, peel angle, peel speed, spring thickness, data region, diameter of drum, and module of spur gear. Peel length is fixed at 60 mm in *x*-axis direction. Control-factor-level is then studied to prevent any misleading results due to incorrect control-factor-level. Final control factor and level is selected in "Act" stage thus activating the "Plan" stage in the next sub-system. Figure 14 summarizes the methodology flow of control measurement system (C):

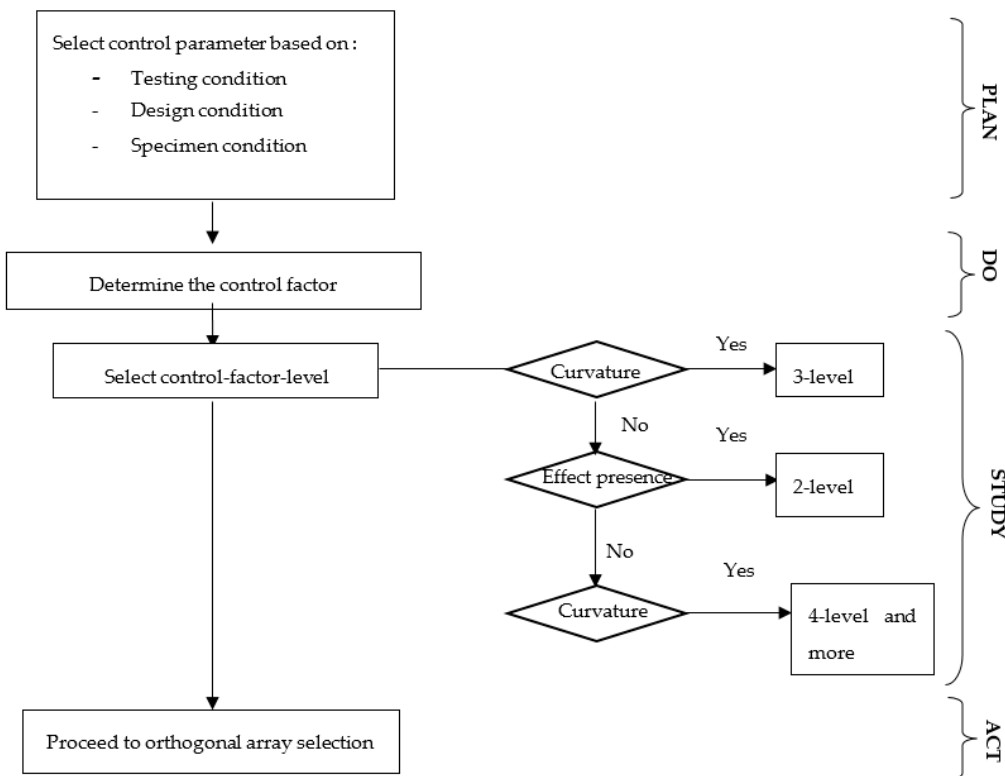

**Figure 14.** Control (C) sub-system of F-N-C-O.

### 4.4. Optimization (O) System

Once the function, noise, and control measurement are done, the optimization stage takes place. "Act" stage in C sub-system acts to activate the "Plan" stage in O sub-system. "Plan" in Optimization sub-system is done by correctly choosing the orthogonal array based on noise and control factor strategies. Orthogonal array is chosen based on number of control factor and noise factor and their levels. Orthogonal array is the design space. It is a balance set of experimentation run. Every pair of columns, all combinations of levels occur in equal number of times. One two-level factor and six three-level control factors are used in L18 that result for 108 experimental runs. Outer array of N1 and N2 with one three-level of signal factor is used. Dynamic SNR is employed. The relationship between the mean response and the levels of signal factor is linear. In "Do" stage, experiment is implemented based on the setting of levels in orthogonal array. The measurement data are recorded and further analysis on SNR is done.

In the "Study" stage, the criticality of data and the assumption is analyzed to ensure the measured data is genuine from extraneous variation that is not in the measurement system. The spread of measured data is checked through linear regression plot for abnormality checking. Next, the confirmation test is done for reproducibility checking. When estimation SNR gain is not comparable with confirmation gain resulting more than 30% difference, the measured data is not reproducible and investigation needs to be done. If this mistake is realized for more than three months, repetition of the experiment of that particular point is needed. However, if the mistake is realized in less than three months, the abnormal data can be replaced with regression point by treating it as missing data. Confirmation experiment is done once again to ensure the reproducibility is less than 30% or 3 dB. Either to repeat or replace with regression point in a linear relation, specimen condition must be considered. This is done to minimize the outer noise and inner noise due to environmental condition and deterioration of elements in the sample, respectively. Thus, the optimum condition is accepted in "Act" stage. Figure 15 summarizes the optimization measurement system:

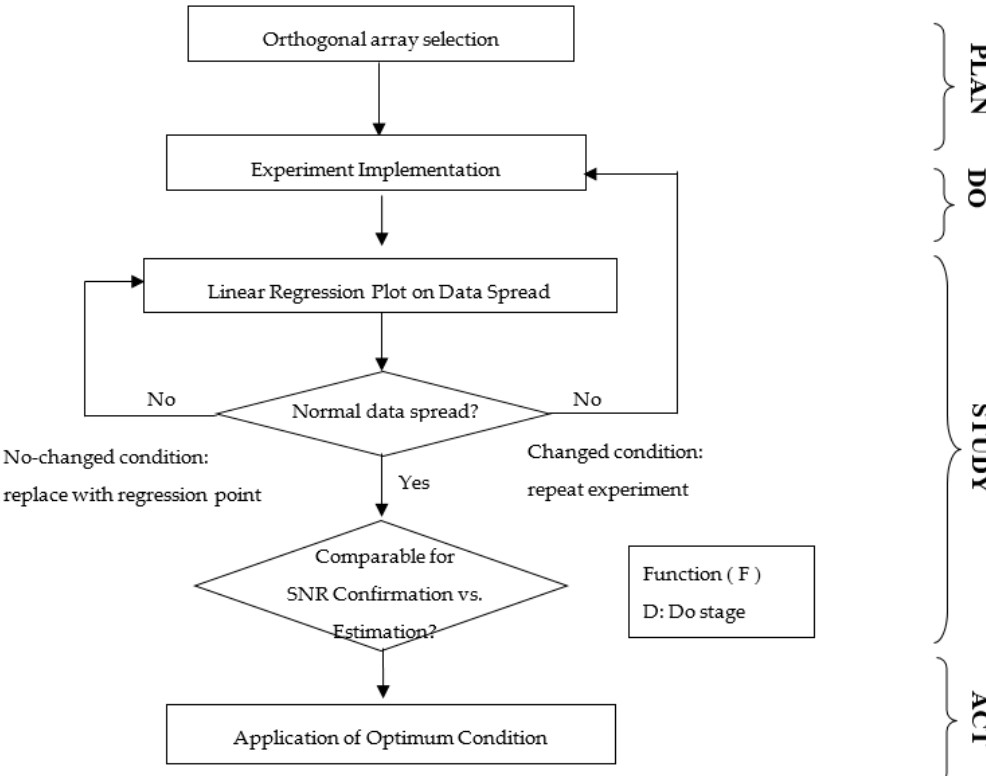

**Figure 15.** Optimization (O) sub-system of F-N-C-O.

## 5. Conclusions

In this study, a robust measurement system using Taguchi Method is developed to improve the standardized T-peel test method using four main subsystems that are F (function), N (noise), C (control), and O (optimization). This is the flow of robust measurement system for parameter design as concluded in Figure 16. While many papers had been focused on application of parameter design, very less concern was on the measurement system behind the optimization result achieved in parameter design implementation. The four sub-systems consisting function—F, noise—N, control—C, and optimization—O describe the flow of measurement system for finding the optimum condition with lowest variation in T-peel test. The new T-peel test device has reduced the variation in the noise factor that is peel angle deviation and variation in peel strength by 71% based on the improvement from worst condition. Experiment is the tool for measurement. The reliability of measurement depends on how well it is planned, how well the data are analyzed, and how the results are evaluated as

discussed in F-N-C-O cycle of measurement systems. The reliability of the optimum condition in the T-peel test is proven reliable as the dB gained is 2.86 dB with 27% difference, which is less than 30%, indicating good reproducibility. The finding of this paper affects the measurement system in Taguchi Method for parameter design by providing higher confidence level and optimization rate. F-N-C-O Taguchi Method presents the theory of experimental design together with the method to achieve the optimum condition. By developing the F-N-C-O Taguchi Method for robust measurement system using robust design engineering method, parameter design becomes more convergent and higher degree of confidence in reliability. It also has enlightened the black box of parameter design of Taguchi method by clarifying the reasons behind optimization result.

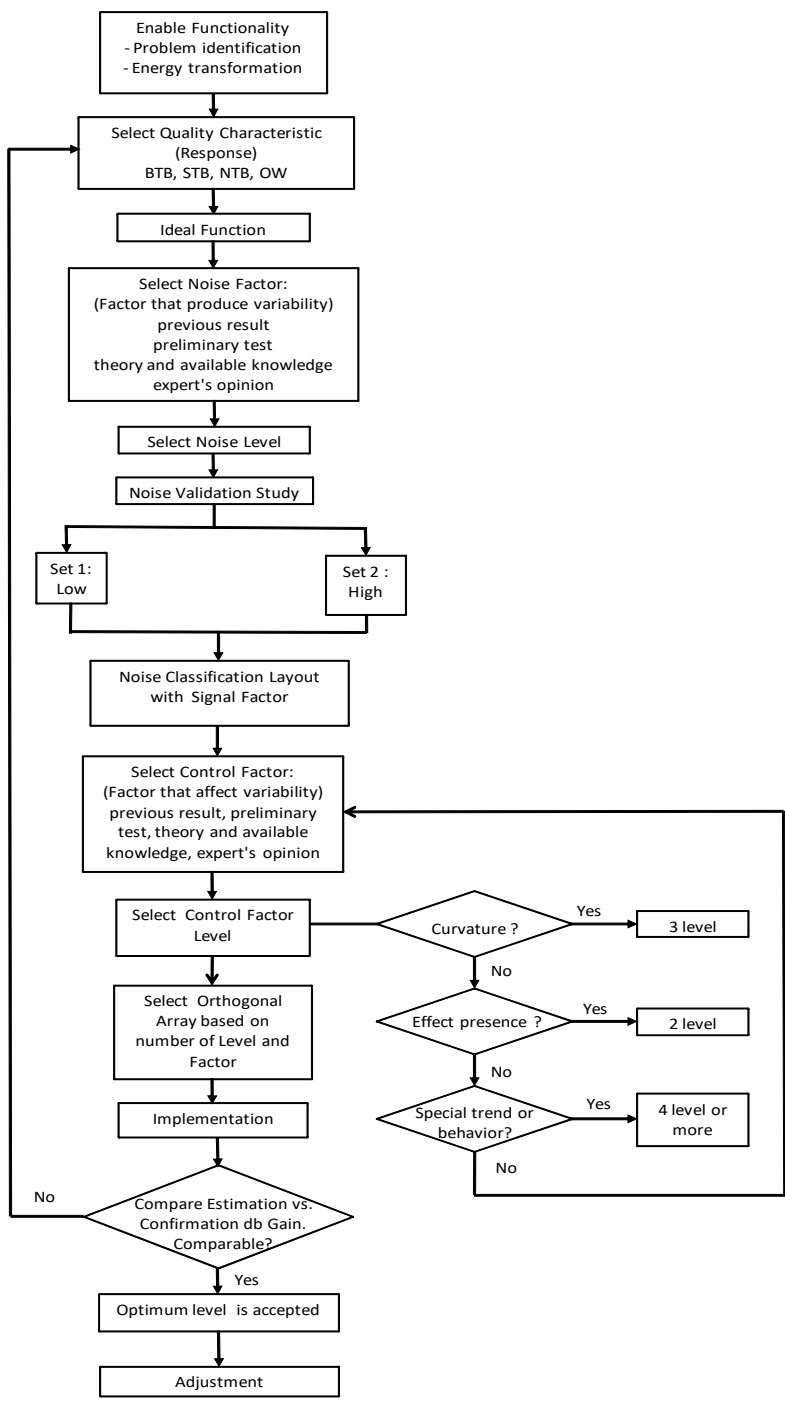

**Figure 16.** Methodology flow for robust measurement system.

**Author Contributions:** A huge appreciation to all authors in producing this manuscript successfully. R.D. as the main author, contributions are in conceptualization, formal analysis of the experimental data, writing the original draft, producing the methodology of F-N-C-O, funding acquisition, and investigation of the optimum solution. To all co-authors: S.K. who contributed in reviewing and editing, validation of result, and conceptualization of this paper. M.Z.H. who developed the methodology and writing the original draft together. M.F.M.D. who supervised the scholar works, visualization, reviewing and editing of the paper. K.R.J. who assisted the formal analysis, visualization of the paper, and reviewing and editing the paper. F.R. who did a great job in administrating the project and investigating the experimental method. All authors have read and agreed to the published version of the manuscript.

**Funding:** This publication was funded by "Geran Universiti Penyelidik" (GUP) Tier 2 Scheme by Universiti Teknologi Malaysia Q.K130000.2656.15J85, Fundamental Research Grant Scheme; FRGS awarded by the Ministry of Higher Education Malaysia (MOHE) for R.K130000.7856.5F247 and R.K130000.7856.5F233, and Malaysia Research University Network (MRUN) Grant R.J130000.7805.4L886.

**Acknowledgments:** The authors fully acknowledge the Malaysia Ministry of Higher Education (MOHE), Meiji University, Japan, and Universiti Teknologi Malaysia (UTM), Malaysia, for the support and scholarship given in making this important research viable and effective. We also would like to thank the industry people from Naglus Industries Sdn. Bhd.—Dato' Suhairi Sa'ad, ASI Consulting Group—Shin Taguchi, Japan's National Institute of Advanced Industrial Science and Technology—Masayoshi Koike, FujiXerox—Kazuo Tatebayashi, Ricoh—Tetsuo Noji, CIGNA Boston—Rajesh Jugulum, ADAAP Process Solutions Pvt. Ltd.—Arun K. Chaudhuri, Angletry Associates—Shoichi Tetshima, and ShinEtsu Handotai Malaysia S.E.H (M) Sdn. Bhd. which have been very cooperative in contributing brilliant ideas and motivation towards data testing, evaluation, measurement method, and metrology field of improvement.

**Conflicts of Interest:** The authors declare no conflict of interest.

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
