# Peer review of "Development of F-N-C-O Taguchi Method for Robust Measurement System Using a Case Study of T-Peel Test on Adhesion Strength"

_applsci, doi:10.3390/app10186203_

Round 1

Reviewer 1 Report

The manuscript deals with the Development of F-N-C-O Taguchi Robust Measurement System using a Case Study of T-Peel Test on Adhesion Strength. The manuscript has a major problem: it was submitted with the track mode on (with many comments included). The abstract does not highlight the strengths of the paper. The introduction does not highlight the relevance of the method being used (T-peel) nor discuss the limitations on the current state-of-the-art with only very general discussion. Some images are missing which indicates poor attention to detail on the submission again. The description of the theory behind the technique is good, but the details regarding experimental set-up are subpar. The Discussions and Conclusions are very general, and should be more focused on the relations to the T-peel test.

Abstract

I believe that this statement "In many literatures, measurement systems have only been reviewed as a test method without explaining the previous stage before the experimental procedure" should not be included in the abstract because it does not cite which are the literatures that do not explain the previous stages. I would start from "In many literatures, measurement systems have only been reviewed as a test method without explaining the previous stage before the experimental procedure" and use the space to focus on the importance of measurements systems.

  1. Introduction

- Please rearrange the phrase to reduce the repetition of the word "robust"

- I believe that would be beneficial to divide the references 15-30 in two groups: (a) the ones who discuss the method and (b) the ones who discuss the theory, instead of citing a lot of references grouping then in a single category "In many literatures, measurement systems have only been reviewed as a test method without explaining the previous stage before the experimental procedure".

- I suggest a much larger discussion of the limitations on the current state of the art, and which are the novelty provided by your study

- Are you sure that your research is the first one to discuss both the method and the theory behind it? It is a very bold statement

- Please discuss the relevance of T-peel method, and making comparisons with the limitations existing in the state-of-the-art.

- Please make clearer the relevance of your research

- Please address the comment already in the document "In many literatures, measurement systems have only been reviewed as a test method

21 without explaining the previous stage before the experimental procedure"

  1. Methodology

- I would rename this section to "State-of-the-art" since it is showing how current things are being done.

- Very good description of the methods

- Figure 4 is missing.

- PLease include an image of the experimental set-up.

  1. Experimental results

- The equations 2 to 4 seem confusing, please recheck.

- Please improve the quality of Figures 6 and 7

  1. Discussions

- The discussion is interesting but is very theory-based, please include more correlation with the experimental results obtained in the present work.

  1. Conclusions

- I would remove this "In many literatures, measurement systems have only been reviewed as a test method without explaining the previous stage before the experimental procedure". The conclusion is the place to make short commentary on your own work. The relation to the state-of-the-art should be done in the introduction. The conclusion is very general. I believe that would be beneficial to focus more on the discussions regarding your own results. How the development of the system affected the T-peel test.

Author Response

2. Methodology

  • State-of-the-Art methodology renamed
  • Figure 4 is added

3. Experiment Results

  • Equation is improved
  • Figure 6 and 7 are improved

4. Some parts have been described together with the experimental condition and results

5. I removed the "In many literatures, measurement systems have only been reviewed as a test method without explaining the previous stage before the experimental procedure". Comments on the current work is made in such a way

I would remove this "In many literatures, measurement systems have only been reviewed as a test method without explaining the previous stage before the experimental procedure". The conclusion is the place to make short commentary on your own work. The relation to the state-of-the-art should be done in the introduction. The conclusion is very general. I believe that would be beneficial to focus more on the discussions regarding your own results. How the development of the system affected the T-peel test.

Reviewer 2 Report

Detailed comments

  1. The font size in the diagrams is too small. It is difficult to read eg. indexes. Please correct it.
  2. Is it necessary to substitute numerical values into formulas and present it in the article, eg. 2-4 ?

Author Response

I have made some enlargement to the diagrams.

Yes, I feel it is necessary for the numerical value in the eq. 2-4 because it will provide better understanding to the reader.

Reviewer 3 Report

It is an interesting topic for investigation. The manuscript is well written. However, in my opinion, the description of FNCO should be moved from the disscussion part to the methodology. For having self-explaining figures and tables, all the caption of figures and tables should be improved.

The table 2 can be removed in the manuscript, because the table 3 showed the matrix of variables as well.

the reason for choosing the 2 or 3 level for each variable can be explained in the methodology.

Author Response

The brief description of FNCO has been introduced in methodology starting line 231.

Table 2 has been removed because Table 3 has explained the orthogonal array. Thus, Table 3 has been labelled as Table 2. 

The reason of choosing the 2 or 3 levels have been described in section 2.2, starting line 178 to 180.

Round 2

Reviewer 1 Report

The authors have followed all suggestions, therefore, I recommend the acceptance of the manuscript.

This manuscript is a resubmission of an earlier submission. The following is a list of the peer review reports and author responses from that submission.

Round 1

Reviewer 1 Report

The paper provides an approach to determine the optimal condition for a measurement system with an attempt to demonstrate the workflow using the T-peel test. However, I am concerned about the quality of the writing as well as the novelty of the article. I think the main idea of this article has been previously published: “Effect of Peel Side on Optimum Condition for Measuring Flexible Film Peel Strength in T-Peel Adhesion Test”, DOI: 10.1520/JTE20120342. I also have the following additional comments that hopefully will help the authors to improve the quality of the paper in the future:

  1. The paper is poorly written with many unnecessary typos and grammar mistakes. For example, even the author list in the title page has typos. First, there is a double commas “,,” in the author list. Second, the affiliations are labeled as “a, b, c”, but then the affiliations are listed with numbers “1, 2, 3”. Overall, the paper is not well written and is hard to follow.
  2. The authors describe the measurement system in the methodology section (lines 73 to 81). Several components are explained, but the role of “spur gear” and “migration stage” are not mentioned.
  3. The authors assess the performance of using noise factor and also mention its variants. However, the authors did not classify the noise factor that was used to evaluate the measurement system. In the line 125, the authors show that peel angle and tensile weight are noise factor, but the tensile weight was classified as the control factor in Figure 3.
  4. More details for control are necessary.
  5. In Table 1, the authors describe the experimental setup that consists of 18 runs x 3 signal levels x 2 noise level. However, there are only two levels mentioned in column A while columns (B – G) have three levels.
  6. In line 78, the authors mention that three levels of spring thickness were used to evaluate the system. However, this feature was not included in the analysis. It is necessary to show how does this feature affects on the peel strength.
  7. Please show the relationship between Table 2 and Figure 7.
  8. Please elaborate on the term optimum and worst condition in Table 3.
  9. The discussions section briefly describes the definition of F-N-C-O but it does not provide a useful discussion to readers.

Reviewer 2 Report

It is a good paper overall with sound methodology. However, needs major restructuring for it to be easy to read. A lot of background information in the methods, results and discussion sections can be moved to the introduction section. There is a lot of repeatability through out the paper. There is no consistency or flow. Special attention is needed for English and writing style. The different writing styles of the different authors is too apparent. The end product must look like one person wrote it. The paper needs more references to make it more, well, robust. Figures are too small or unclear. All necessary terminology must be clearly defined in the introduction section. 

Line 20: has been

Line 21: The gap...Revise sentence

Line 32: and gains

Line 40: Give references for the case studies and for papers that only discussed measurement concepts

Line 43: measurement are described

Materials section is missing: Where did you get the tape? Is that standard 

The Methods section needs major reorganization. Maybe the steps at the end of the section can go first. You tell the reader what you are doing. Explain the method and then demonstrate.

You talk about multiple flexible packaging? But later only Al-CPP is mentioned.

Line 63: Thus, a peel test....with adhesives (Not clear what you are saying)

Fig 1 needs to be cleared. Its too pixelated. Difficult to see how the tape is attached to the drums

Line 89-93: More suitable in the introduction

Line 94: All the parameters not "whole parameters"

Line 96: which is measured in Newton (N)

Line 122: What historical data? References

Line 136: What historical data? References. Who are these experts in your case?

Figure 4: Which way is +2 and -2?

Figure 3: Label A-G

Figure 7: Needs to be bigger

Equation 2, 3, and 4: Define all the variables. 

Line 257: Remove "the"

Line 295: Variable that affects

Line 331-332: Same thing twice

Line 340-355: Belongs in the Introduction section

Line 371-380: How you decided L18 must be in the Methods section. Why are you suddenly talking about L9? Did you show us the results of L9? Did you mean to show us L9 but forgot?

Explain briefly in the introduction section what is the Taguchi method. And why you chose it?

Figure 13: Where is the Yes? Figure needs to be bigger. The text is too small.